

# High expression of MAPK-14 promoting the death of chondrocytes is an important signal of osteoarthritis process

Zhiqin Deng[1,*], Xiaotian Hu[1,2,*], Murad Alahdal[1], Jianquan Liu[1], Zhe Zhao[1], Xiaoqiang Chen[1], Junxiong Xie[1], Li Duan[1], Daping Wang[1,2] and Wencui Li[1]

[1] Hand and Foot Surgery Department, Shenzhen Second People's Hospital (The First Hospital Affiliated to Shenzhen University), Shenzhen, Guangdong, China
[2] Anhui Medical University, Hefei, China
[*] These authors contributed equally to this work.

Corresponding authors
Daping Wang,
wangdp@mail.sustech.edu.cn
Wencui Li, 13923750767@163.com

## ABSTRACT

**Background.** Osteoarthritis (OA) is one of the most common degenerative diseases worldwide. Many researchers are studying the pathogenesis of OA, however, it is still unclear.

**Methods.** Screening and validation of OA relevant hub genes are an important part of exploring their potential molecular mechanism. Therefore, this study aims to explore and verify the mechanisms of hub genes in the OA by bioinformatics, qPCR, fluorescence and propidium iodide staining.

**Results.** Microarray datasets GSE43923, GSE55457 and GSE12021 were collected in the Gene Expression Omnibus (GEO), including 45 samples, which divided into 23 osteoarthritis knee joint samples and 22 samples of normal knee joint. Thereafter, 265 differentiallyexpressedgenes (DEGs) were identified in all, which divided into 199 upregulated genes and 66 downregulated genes. The hub genes MAPK-14, PTPRC, PTPN12 were upregulated, while B9D1 was downregulated. In order to further confirm the expression of screening differential genes in human chondrocytes, the human chondrocytes were extracted from a joint replacement surgery and stained with toluidine blue for identification. Compared with normal chondrocytes, OA chondrocytes had high expression of COL I protein and low expression of COL II protein. The expression levels of MAPK-14, PTPRC and PTPN12 in OA chondrocytes were significantly higher than the expression levels of B9D1 in normal chondrocytes. Moreover, the inflammatory necrosis of OA chondrocytes was increased compared with the normal chondrocytes by propidium iodide staining.

**Conclusions.** The high expression of MAPK-14 works as a promoter of chondrocytes death and an important signal of the osteoarthritis process.

Subjects Bioinformatics, Cell Biology, Orthopedics, Rheumatology
Keywords Osteoarthritis, MAPK-14, Chondrocytes, Differentially expressed genes, Protein-protein interaction, Gene Ontology

## INTRODUCTION

Osteoarthritis is the most serious chronic bone disease and can cause severe pain and even disability in the elderly (*Song, Chang & Dunlop, 2006*). OA not only brings suffering to patients but also aggravates their economic burden. Nearly half of the elderly patients are receiving treatment, and most of them are still suffering or disabled (*McKenzie & Torkington, 2010*). About one-third of elderly over 60 years old are supposed to be diagnosed with OA by 2030 (*Croft, 2005*). One of the most important risks in OA development is the pathogenic changes of hub genes (*Hochberg et al., 2013*). Heritability of osteoarthritis reached approximately 50% (*Cibrian Uhalte et al., 2017*). However, the molecular mechanism genetic changes in OA remains unknown.

However, current researches showed some disadvantages such as using small amount of genetic samples and the lack of relevant experimental verification. Therefore, the expanding of the number of screening sample size on the basis of the predecessors, optimizing the relevant screening conditions and carrying out the relevant verification through cell lines experiments could contribute to further understanding and enhances further study of the internal mechanism regarding the occurrence and development of osteoarthritis. Recently, microarray technology and bioinformatics have made significant advances which contribute to understand differentially expressed genes (DEGs) in OA and enable us to do further studies. Here, we obtained three datasets from Gene Expression Omnibus (GEO) for analysis and obtain DEGs between OA knee joint samples and normal knee joint samples. Based on this, we not only analyzed Gene Ontology (GO), Kyoto Encyclopedia of Genes and Genomes (KEGG) pathway enrichment of DEGs, but also produced protein-protein interaction (PPI) network to help us unlock some of the unstudied parts of the molecular mechanisms of OA. The purpose of this research was to explore regarding underlying mechanisms of OA, so as to promote the further development of OA treatment.

## MATERIALS & METHODS

### Microarray data information

The Gene Expression Omnibus (GEO, http://www.ncbi.nlm.nih.gov/geo) is a genomic database for all users, from which users can get all kinds of data free of charge (*Barrett et al., 2013*).The expression profiles of GSE43923, GSE55457 and GSE12021 were all searched and got from the GEO. GSE43923 include 3 OA knee joint samples and 3 normal knee joint samples. GSE12021 include 10 OA knee joint samples and 9 normal knee joint samples, and GSE55457 include 10 OA knee joint samples and 10 normal knee joint samples.

### Identification of DEGs

GEO2R (http://www.ncbi.nlm.nih.gov/geo/geo2r), a free public network platform, was used to process the DEGs between OA knee joint samples and normal knee joint samples. Using GEO2R, we have integrated and analyzed multiple datasets for identifying DEGs. $|logFC| > 1$ and $P$-value $< 0.05$ were considered the cutoff criterion and have statistically significant.

## GO and KEGG enrichment analyses of DEGs

DAVID (http://david.ncifcrf.gov), a free public online biological information database was used to provide biological function analysis for a large number of genes and can visualize the results (*Huang et al., 2007*). GO is a biological information analysis tool that can induce and analyze a large number of genes and visually export corresponding results, including BP, CC, and MF (*Shannon et al., 2003*). KEGG is a utility database resource for understanding advanced functions and biological systems (such as cells, organisms, and ecosystems), generating genomic sequencing and other high-throughput experimental technologies from molecular level information, especially large molecular datasets (*Szklarczyk et al., 2011*). Deep analyses were implemented by using the DAVID online database. $P < 0.05$ was considered statistically significant.

## PPI network construction and module analysis

STRING (https://string-db.org/cgi/input.pl), a free public online search platform was used to elucidate the relationship between interacting genes (*Franceschini et al., 2013*). DEGs previously screened in this research were imported into STRING to product the PPI network, and interaction with a confidence score >0.4 was considered significant. Cytoscape, a free and public source bioinformatics application that we imported the data into it. The built-in APP Molecular Complex Detection (MCODE) of Cytoscape is convenient to select the most significant modules in all DEGs and identify the most prominent gene in each module (*Bader & Hogue, 2003*).

## Isolation and culture of chondrocytes

Cartilage specimens of the fracture patients' femoral head and the knee joint of patients with OA in Shenzhen second people's hospital were collected to isolate and cultivate chondrocytes. The adherent cells were cultured in the chondrocyte medium, which contains the DMEM-LG (Gibco), 10%FBS (Gibco), 1% penicillin-streptomycin (Gibco), 1%HEPES, 1% vitamin solution, 1% proline solution and 1% non-essential amino acid NEAA. These adherent cells are what we call chondrocytes. chondrocytes were cultured in at 37 °C and 5% $CO_2$. The study was reviewed and approved by the First Affiliated Hospital of Shenzhen University Health Science Center Research Ethics Committee. All patients signed a written informed consent before participating in this study.

## Toluidine blue staining of chondrocytes

First, we washed the Petri dishes with PBS for 2 times and fixed the cells with 4% paraformaldehyde directly for about 0.5 h (4 °C). Then wash with tap water for 15 min, then with distilled water for 5 min, then add 1% toluidine blue for 2 h. Finally, remove excess dye and filter impurities. The dyeing condition was observed under the microscope, dried after satisfaction, and sealed with neutral gum.

## Immunofluorescence staining of chondrocytes

PBS washed the cells twice. The cells were fixed with paraformaldehyde for 15 min and washed with PBS for 6 times/5min. 0.5% Triton-X100 was incubated and drilled for 15 min. Wash PBS 6 times/5 min.The cells were sealed with 5% BSA at room temperature for

2 h and cleaned with 0.5% BSA for 6 times/5 min. The primary antibody (Rabbit Anti-Collagen II/FITC Conjugated antibody (bs-10589R-FITC) and Rabbit Anti-Collagen I/Cy3 Conjugated antibody (bs-10423R-Cy3), Bioss, Beijing, China) were added and incubated at room temperature for 2 h and washed with PBS for 6 times/5 min. DAPI staining was added for 5min. Photos were taken using a confocal microscope ZEIZZ LSM800, Germany.

## Western blot

OA chondrocytes and normal control were cultured for three days, and then it was taken out of the $CO_2$ incubator, cells were washed twice with PBS. Total protein was obtained by lysis buffer (Beyotime, China), and then placed on ice for 30 min. Centrifuged the sample at 12,000 g for 25 min at 4 °C. The supernatants were transferred to a new EP tube. The protein concentration was measured by BCA quantitative method. Protein samples were electrophoresed on 10% SDS-PAGE gel. After that, protein samples were transferred on PVDF membranes according to the three-character sandwich method. Use 5% skimmed milk powder to block for 2 h. Use the primary antibody containing COL I (1:1,000, boss, China), COL II (1:1,000, Thermo Scientific, USA) and GAPDH (1:1,000, abcam, USA) to co-incubate with the sample overnight at 4 °C. TBST cleaning, 5 times/10 min. Then use the second antibody containing goat anti-rabbit incubation bath for two hours at room temperature. After washing the unbound antibody, use chemiluminescence to take pictures in the two-color laser and chemiluminescence imaging system (Odyssey FC, Gene Company Limited, China).

## Apoptosis assay of chondrocytes

The chondrocytes were isolated from the cartilage tissue, washed and then resuspended, and the precipitate was taken. About 50 uL cell suspension was added into each flap-throwing tank and centrifuged in a flap-throwing centrifuge (1,500 g, 10 min). The chondrocytes can be attached to a slide after the surrounding water is absorbed by the filter paper. Then wash with PBS for 3 times and fix with 4% paraformaldehyde for 15 min. Wash again with PBS for 3 times, add PI immunofluorescence staining and incubate in a wet tank for 30 min. After washing with PBS again for 3 times, fluorescence microscope observation and image collection were performed.

## qRT-PCR validation and statistical analysis

After previous research, we've found four hub genes, including MAPK14, PTPRC, B9D1, PTPN12. We searched the primers of the four hub genes through the PrimerBank (https://pga.mgh.harvard.edu/primerbank/) and ordered the relevant primers through the Sangon Biotech. Quantitative real-time polymerase chain reaction primers are detailed in Table 1. Total RNA was extracted from chondrocytes using the RNeasy mini kit (Takara), according to the manufacturer's instructions. The RNA was reversely transcribed into cDNA by iScriptTM cDNA Synthesis Kit (Takara) and qRT-PCR was performed with a Real-time quantitative PCR instrument (Life Technologies).

## Statistical analysis

Use SigmaPlot 10.0 to draw. Use IBM SPSS Statistics 19 to conduct statistical analysis, data are expressed as mean ± std. A significant statistical difference was $P < 0.05$.

**Table 1  The primers of four hub genes.**

| Gene | Primer sequence | Annealing Temperature (°C) | Length | PrimerBank ID |
|------|-----------------|----------------------------|--------|---------------|
| MAPK14 | Fwd 5′-CCCGAGCGTTACCAGAACC-3′ | 62.4 | 19 | 194578904c1 |
|  | Rev 5′-TCGCATGAATGATGGACTGAAAT-3′ | 60.4 | 23 |  |
| PTPRC | Fwd 5′-ACCACAAGTTTACTAACGCAAGT-3′ | 60.4 | 23 | 18641362a1 |
|  | Rev 5′-TTTGAGGGGGATTCCAGGTAAT-3 | 60.5 | 22 |  |
| B9D1 | Fwd 5′-TGGAGGAGGGGATCTCACAG-3′ | 61.9 | 20 | 343478278c1 |
|  | Rev 5′-CCGTAGGGGTTGGTGCTTTT-3 | 62.4 | 20 |  |
| PTPN12 | Fwd 5′-AGTTGCCTTGTTGAAGGGGAT-3′ | 61.6 | 21 | 196114950c1 |
|  | Rev 5′-AGAAGGTGTCAAGATGGGTGG-3 | 61.4 | 21 |  |

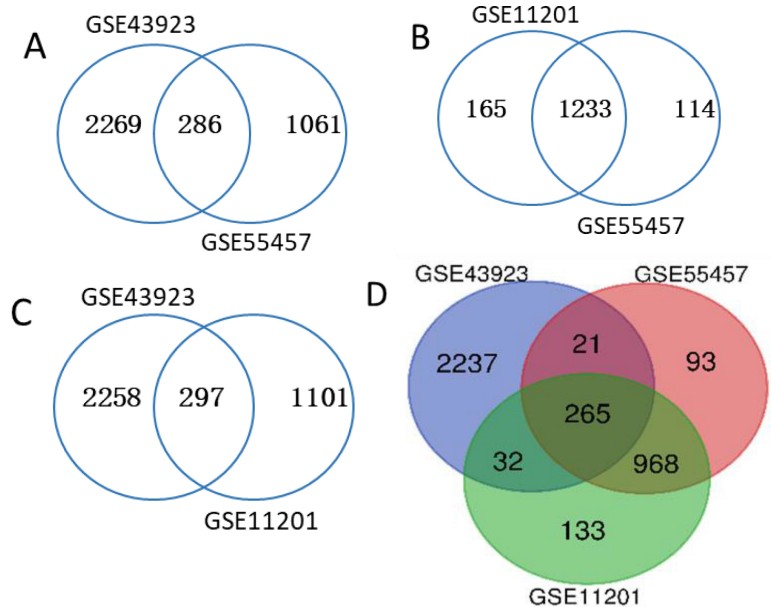

**Figure 1  Venn diagram.** (A) GSE43923 and GSE55457. (B) GSE11201 and GSE55457. (C) GSE43923 and GSE11201. (D) GSE43923, GSE55457 and GSE11201.

## RESULTS

### Identification of DEGs in osteoarthritis

A total of 23 OA knee joint samples and 22 matched normal knee joint samples were analyzed; taking $P < 0.05$ and $|logFC| > 1$ as a threshold, DEGs (2555 in GSE43923, 1347 in GSE55457 and 1398 in GSE11201) were identified. First, we made a pairwise comparison and got three Venn graphs (Figs. 1A, 1B and 1C). Then we comprehensively compared and plotted the final Venn diagram of the differentially expressed gene (Fig. 1D). A total of 265 common differential genes were screened, including 66 down-regulated genes and 199 up-regulated genes.

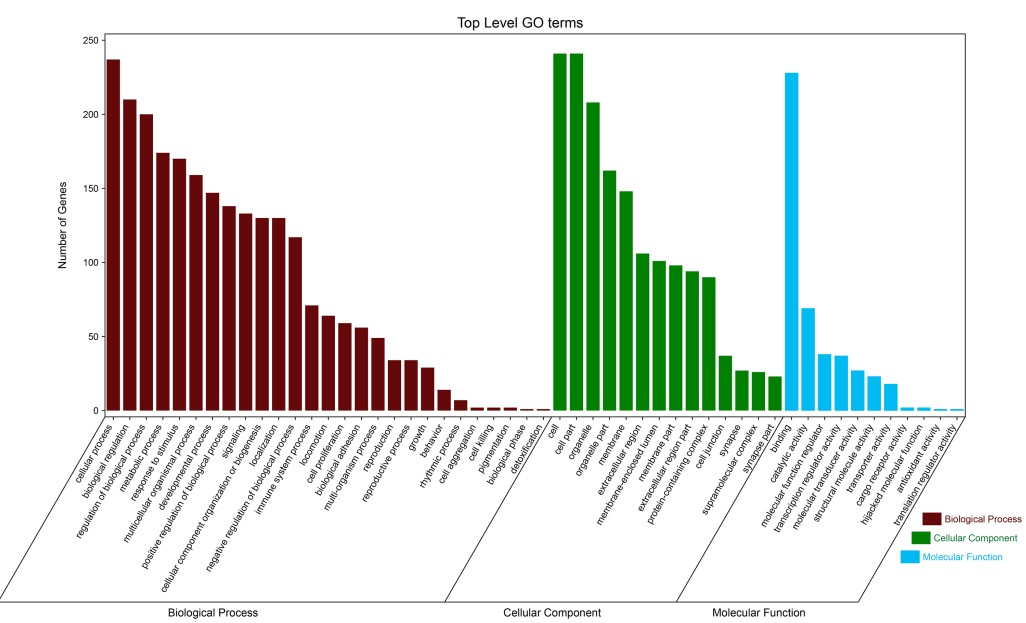

**Figure 2** Gene Ontology (GO) analysis classified the DEGs into three groups: molecular function, biological process, and cellular component.

## KEGG and GO enrichment analyses of DEGs

By using DAVID, the DEGs functions were classified into 3 main categories, including BP, MF, and CC (Fig. 2). We can see that otherness in BP were significantly enriched in anatomical structure formation involved in morphogenesis, cell adhesion, vasculature development, blood vessel development (Fig. 3A). Otherness in MF was mainly enriched in calcium channel inhibitor activity, signaling receptor binding, growth factor activity, cytoskeletal protein binding and adenylate cyclase activator activity (Fig. 3B). Otherness in CC was mainly enriched in the cytoplasmic part, cytoplasm and extracellular region (Fig. 3C).

## PPI network construction and module analysis

We use the data obtained through STRING and then build the PPI network through Cytoscape, containing 140 nodes and 356 edges (Fig. 4). Next, we identified 4 hub genes, including MAPK14, PTPN12, PTPRC and B9D1. Then we analyzed the 4 modules using MCODE (the plug-in for Cytoscape), the four modules were shown in (Fig. 5). Functional enrichment analyses were also carried out for these modules. Pathway enrichment analysis showed that Module 1 is the signaling pathways regulating pluripotency of stem cells and mainly associated with leukocyte transendothelial migration. Module 2 is mainly associated with the regulation of actin polymerization or depolymerization. Module 3 is mainly associated with the positive regulation of response to external stimulus and leukocyte migration. It also can response to lipopolysaccharide. Module 4 is mainly associated with the docking of the ciliary basal body-plasma membrane (Table 2).

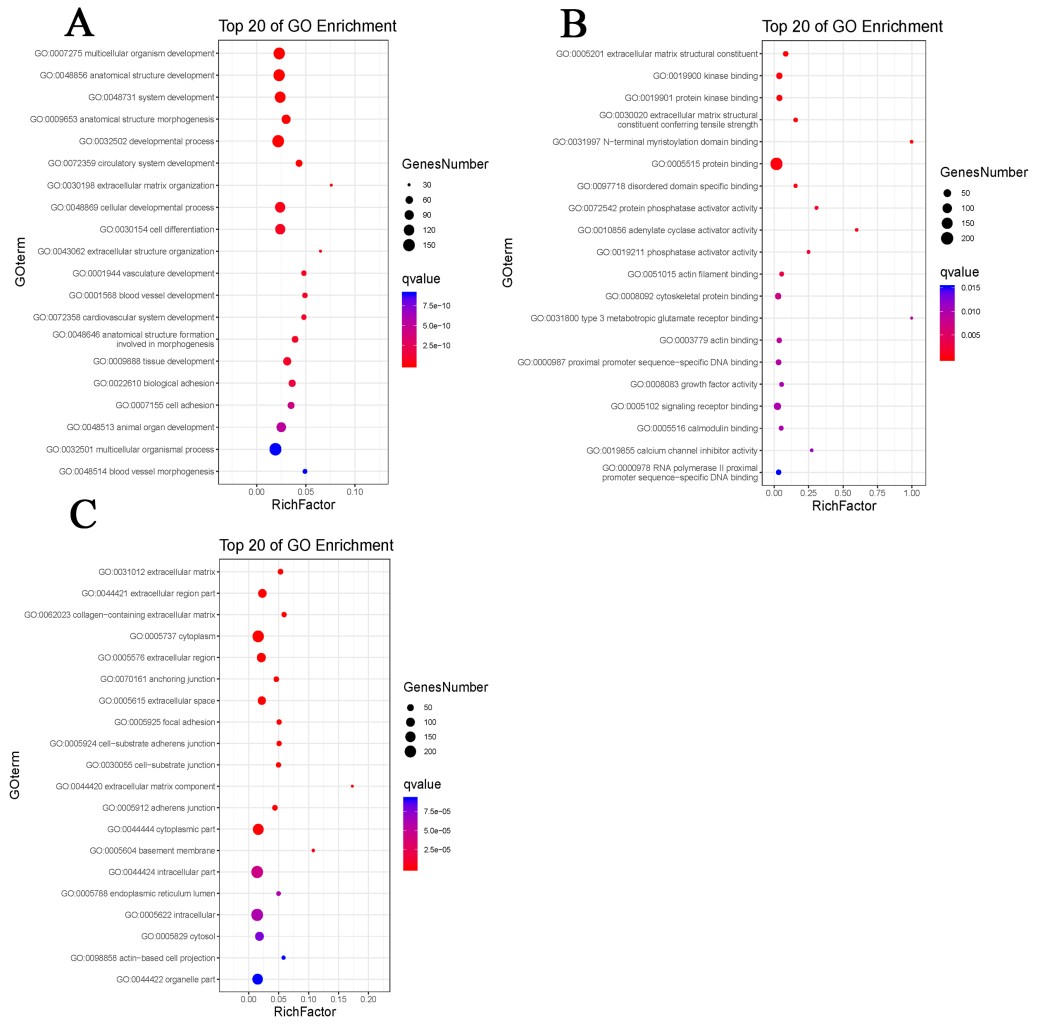

**Figure 3** **Gene Ontology (GO) enrichment analysis.** (A) Enrichment analysis of biological processes (BP). (B) Enrichment analysis of molecular function (MF). (C) Enrichment analysis of cellular component (CC). The gradual color represents the *P* value; the size of the black spots represents the gene number.

## Morphological observation and authenticate of normal chondrocytes and OA chondrocytes

When chondrocytes are completely adherent to the dish, they are mainly flat and polygonal. The morphology of OA chondrocytes was more irregular, with rare cell community aggregation (Fig. 6A).

The cytoplasm of chondrocytes stained with toluidine blue was blue (Fig. 6B). The results showed that all the cells we cultured were chondrocytes.

## Biological identification of OA and control chondrocytes

COL I, COL II and DAPI immunohistochemical tested were carried out on the Chondrocytes. And then we looked at it through a fluorescence microscope, the nuclei

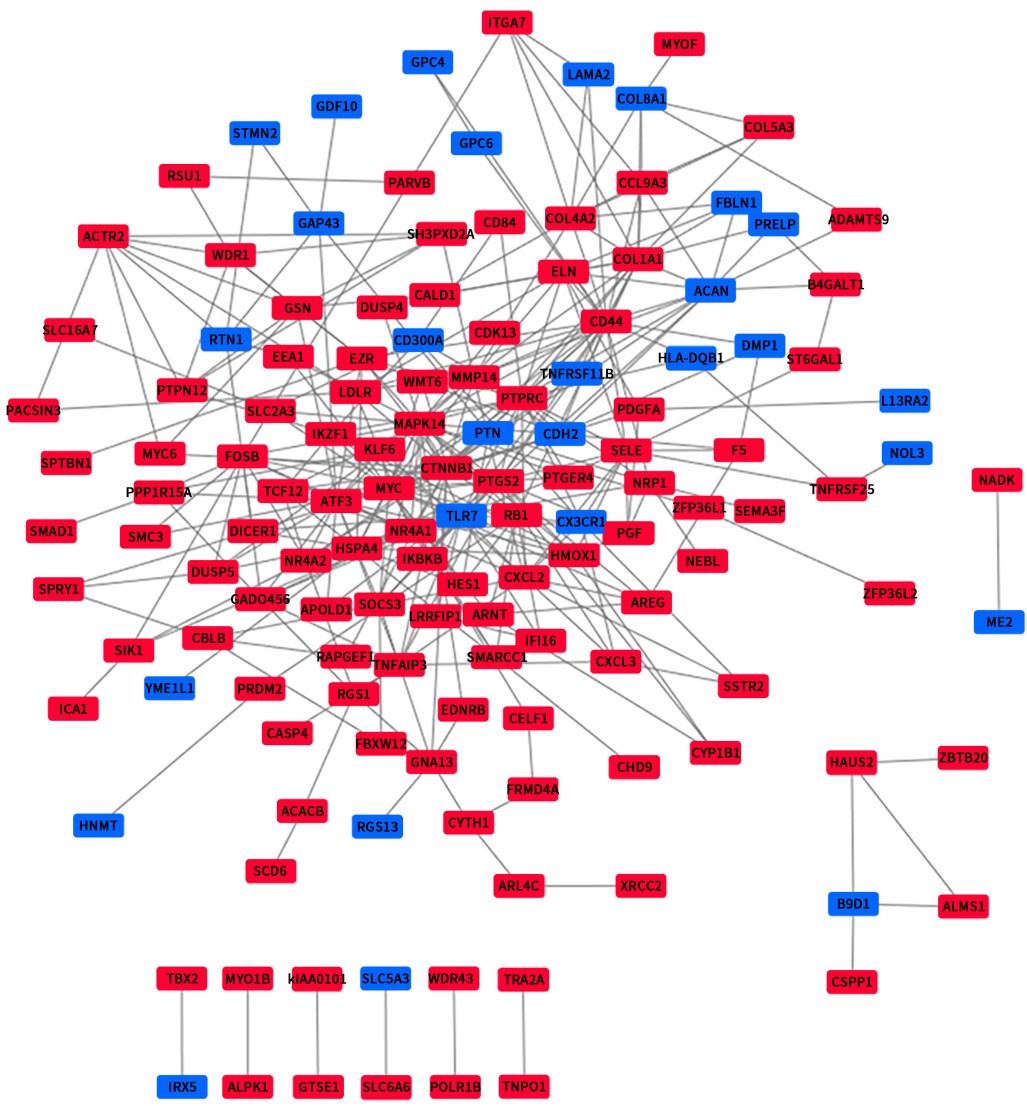

**Figure 4** **The PPI network of DEGs.** Upregulated genes are marked in red; downregulated genes are marked in blue.

are dyed blue, collagen type I dyed red, collagen type II dyed green. Results showed that OA chondrocytes compared with normal chondrocytes, the COL I expression increased (Figs. 6C–6H and 6O), and the expression of COL II reduced (Figs. 6I–6N and 6O) detected by immunofluorescence and westernbolt. This indicates that OA chondrocytes mainly express collagen type I, which means that fibrochondrocytes are newly formed on the surface of cartilage, and the spontaneous depolymerization of fibrochondrocytes further accelerates the death of chondrocytes.

## Validation of hub gene

4 hub genes have been found. Their details are shown in Table 3. We validated our results by measuring the relative expression of our hub genes using qRT-PCR. As can

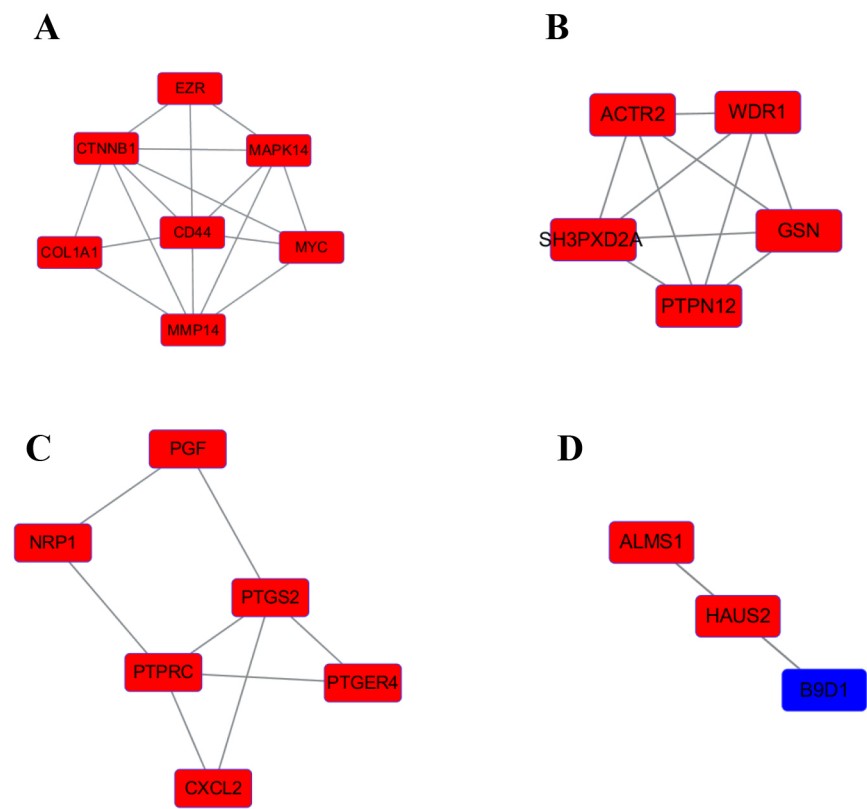

**Figure 5** **The four significant modules were obtained from PPI network.** Upregulated genes are marked in red; downregulated genes are marked in blue. (A) MAPK14 module. (B) PTPN12 module. (C) PTPRC module. (D) B9D1 module.

**Table 2** **Signaling pathway enrichment analysis of hub DEGs function in OA.**

| Expression | Term | Description | Gene Count | LogP |
|---|---|---|---|---|
| UP-DEGs | hsa04550 | Signaling pathways regulating pluripotency of stem cells | 3 | −5.43 |
| | hsa04670 | Leukocyte transendothelial migration | 3 | −5.69 |
| | GO:0008064 | Regulation of actin polymerization or depolymerization | 3 | −5.4 |
| | GO:0032103 | Positive regulation of response to external stimulus | 3 | −6.33 |
| | GO:0097529 | Myeloid leukocyte migration | 3 | −4.88 |
| | GO:0032496 | Response to lipopolysaccharide | 3 | −4.31 |
| DOWN-DEGs | GO:0097711 | Ciliary basal body-plasma membrane docking | 3 | −7.23 |

be seen, the relative expression of MAPK-14, PTPN12 and PTPRC in OA chondrocytes were significantly increased, while the expression of B9D1 were significantly decreased ($P < 0.05$) (Figs. 7A–7D), which was consistent with our expected results.

## Apoptosis test of chondrocytes

The chondrocytes were stained with iodide propidium (PI) and examined by fluorescence microscope. Results showed that compared with normal chondrocytes, a large number

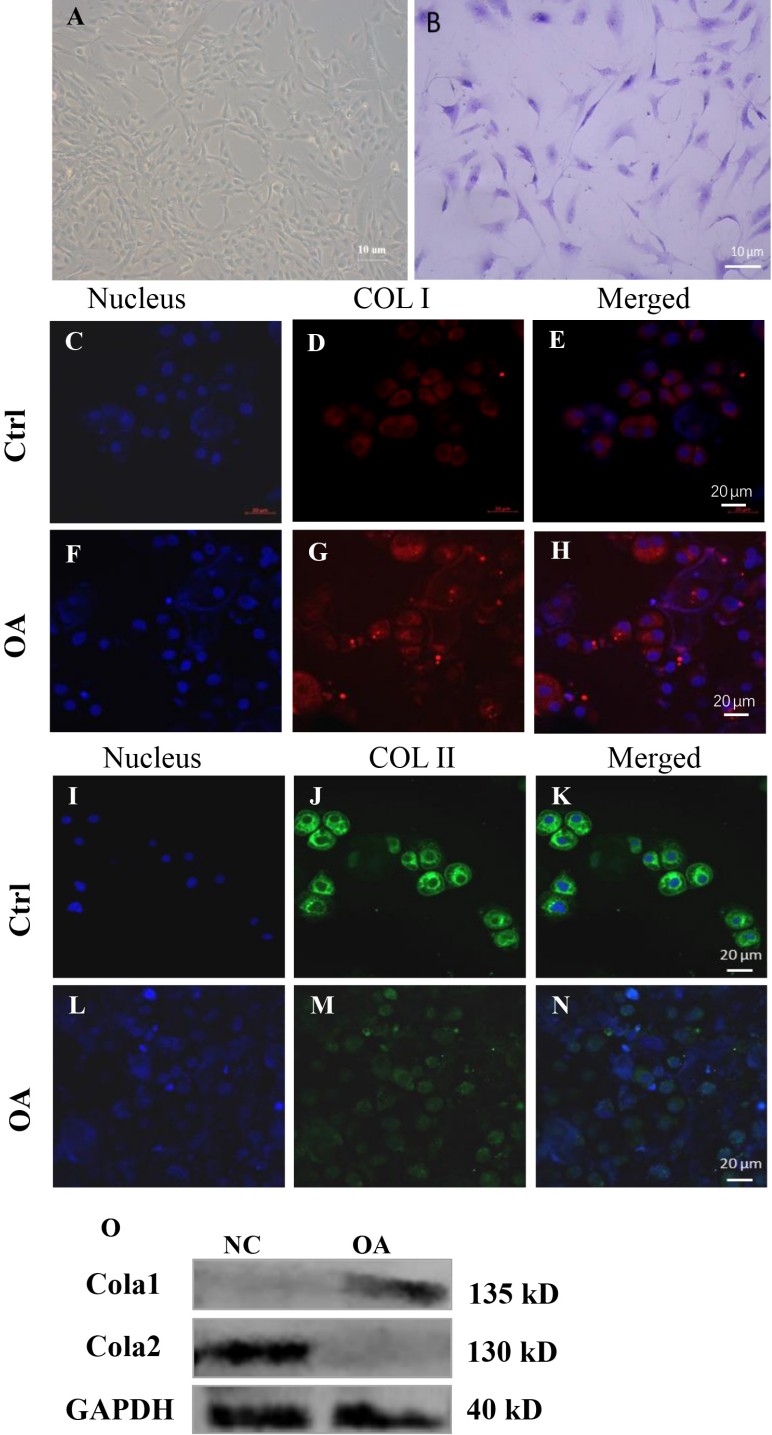

**Figure 6  Cell biology detection.** (A) Morphological observation of chondrocytes. (B) Microscopic observation of toluidine blue staining of chondrocytes. (C)–(N) are the expression of COL I and COL II by immunofluorescence staining in human chondrocytes. (O) is the expression of COL I and COL II by fluorescence western blot in human chondrocytes.

**Table 3  Functional roles of four hub genes.**

| No. | Gene symbol | Full name | Function |
|-----|-------------|-----------|----------|
| 1 | MAPK-14 | Mitogen-activated protein kinase 14 | Play an important role in the cascades of cellular responses evoked and inhibit the lysosomal degradation pathway of autophagy |
| 2 | PTPRC | Protein tyrosine phosphatase receptor type C | High expression of PTPRC is associated with progression of PTP family |
| 3 | PTPN12 | Protein tyrosine phosphatase non-receptor type 12 | High expression of PTPN12 is associated with progression of PTP family |
| 4 | B9D1 | B9 domain containing 1 | Component of the tectonic-like complex and acting as a barrier that prevents diffusion of transmembrane proteins between the cilia and plasma membranes |

of OA chondrocytes were surrounded by pathological vesicles and showed intracellular red staining, which was the PI entering the cells (Figs. 7E–7J). It suggested that the cell membrane was incomplete and the cells death of OA chondrocytes was increased.

## DISCUSSION

The incidence of OA is a second after heart diseases in the statistical reports of some countries. OA, cardiovascular, cerebrovascular diseases and cancer are also called "three major killers" threatening human health. The most immediate symptom of OA is joint pain (*Torres et al., 2006*). A large number of epidemiological studies have confirmed that osteoarthritis is a complex disease involving multiple factors (*Neogi & Zhang, 2013*). Therefore, the molecular mechanisms of OA are the most important direction we can study. It can help us to make early diagnosis and treatment of OA.

In previous researches on bioinformatic analysis, many of them are only limited in two cohort studies or two genetic events. This study makes up for the deficiency in this aspect, we used 3 datasets to obtain DEGs between OA knee joint samples and normal knee joint samples. This can reduce the contingency of analysis. 265 DEGs were identified in all, which divided into 199 upregulated genes and 66 downregulated genes.

GO and KEGG enrichment analyses were performed found that the upregulated genes were mainly enriched in calcium channel inhibitor activity, cytoskeletal protein binding, adenylate cyclase activator activity, while the downregulated genes were mainly enriched in vasculature development, and blood vessel development. According to previous studies, calcium channels have been shown to mediate the OA pain pathway (*Rahman, Patel & Dickenson, 2015*). In addition, some studies have confirmed that the cytoskeletal structure is incomplete and diseased in cartilage chondrocytes of OA (*Blain, 2009*). Other researches had indicated a protective role for pituitary adenylate cyclase-activating polypeptide (PACAP) in chondrocyte differentiation and metabolism (*Strange-Vognsen, Arnbjerg & Hannibal, 1997*; *Juhasz et al., 2014*; *Juhasz et al., 2015*; *Juhasz et al., 2015*). In addition, some studies have pointed out that one of the most important members of osteogenesis is the skeletal vasculature and distinct blood vessel subtypes play different roles in osteogenesis (*Sivan, De Angelis & Kusumbe, 2019*; *Cui et al., 2016*). Moreover, the enriched KEGG pathways of DEGs and modules analysis included NF-kappa B signaling pathway, TNF

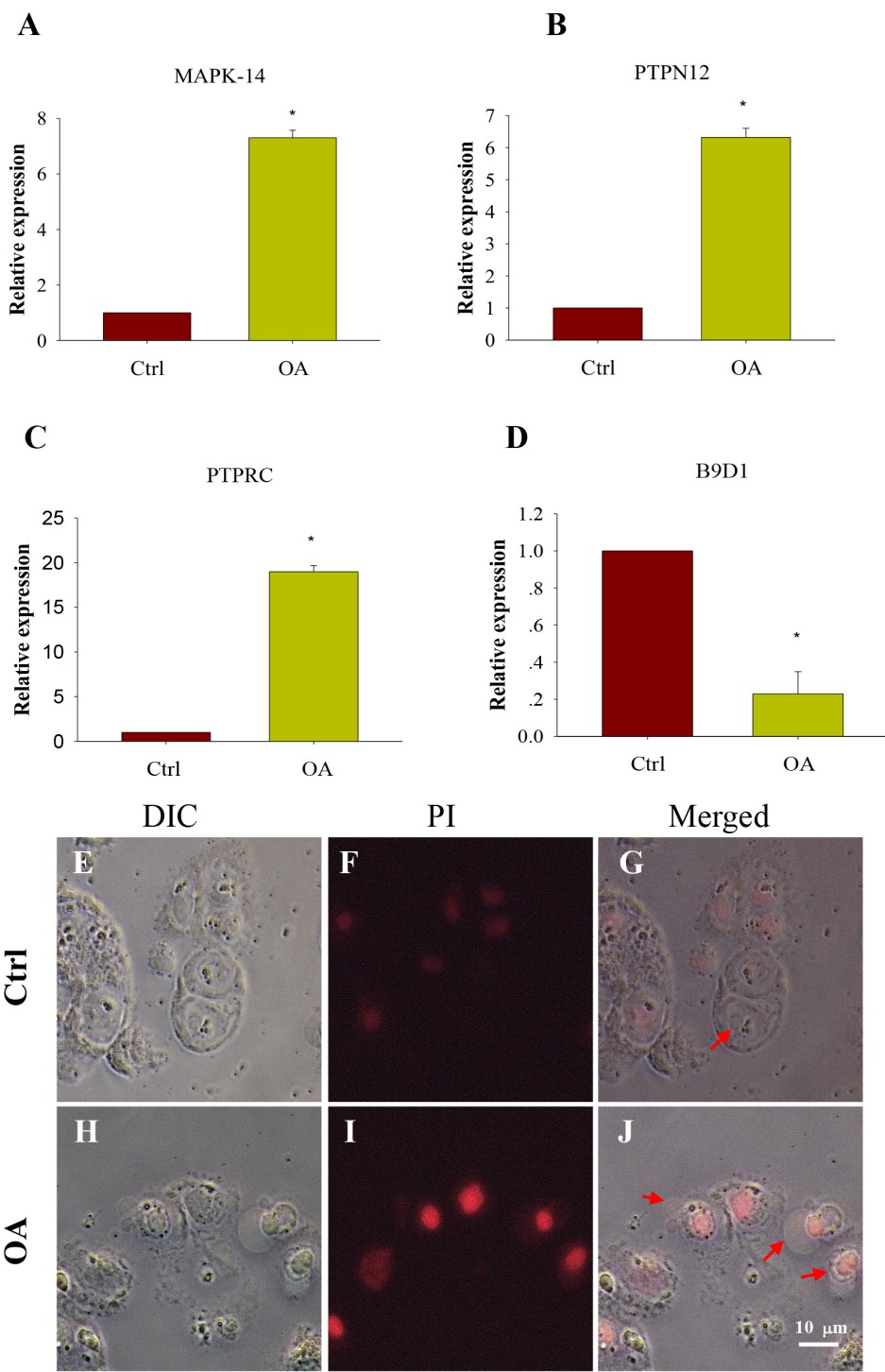

**Figure 7** **Comparison of expression of potential key genes and Apoptosis test in chondrocytes.** (A. MAPK-14; B. PTPRC; C. PTPN12; D. B9D1). All samples were normalized to the expression of GAPDH, and the relative expression levels of each gene were analyzed using the $2^{\Delta\Delta CT}$ method. (E–J) Apoptosis test of chondrocytes (DIC, bright field imaging; PI, Propidium iodide staining imaging; Merged, composite imaging). The red arrow refers to cells that are stained or not by PI.

signaling pathway, PI3K-Akt signaling pathway and MAPK signaling pathway. Previous studies showed that PI3K-Akt,NF-kappa B and TNF signaling pathways are involved in synovial hyperplasia and osteoarthritis cartilage degeneration (*Fu et al., 2016*; *Lai et al., 2014*) and changes in the expression of cytokines, biological factors and enzymes on chondrocytes in human osteoarthritis may be caused by the p38-MAPK signaling pathway (*Feng et al., 2017*). These theories are also consistent with our results.

In this study, normal chondrocytes and OA chondrocytes were extracted from the collected cartilage specimens of the knee joint of patients with femoral head and OA. The cytoplasm of chondrocytes stained with toluidine blue was purplish blue. We found that the expression levels of MAPK-14, PTPN12 and PTPRC in OA chondrocytes were significantly increased, while the expression levels of B9D1 were significantly decreased by qPCR test. This indicates that the changes in the expression level of the potential key genes in the development of OA are consistent with the changes in the expression level of OA chondrocytes, which is consistent with our conjecture.

However, does the high expression of MAPK-14 predict chondrocyte death? The chondrocyte apoptosis experiment showed that the cell membrane of OA chondrocytes was more incomplete than that of normal chondrocytes, and pathological vesicles appeared, indicating that the amount of necrosis of OA chondrocytes was significantly increased. It has been proved that chondrocytes are doomed to die in the development of OA, while high expression of MAPK-14 plays a role in inducing the autophagy of cells.

MAPK14 is one of the representatives of the MAPK family. Furthermore, the MAPK signaling pathway is the most important signaling system to mediate osteoarthritic cartilage damage (*Chowdhury et al., 2008*). Studies have shown that the major pathological changes of osteoarthritis are degradation of the extracellular matrix, leading to progressive loss of cartilage components and destruction of chondrocyte structure and function (*Stoddart et al., 2009*). Inflammatory factors and growth factors specifically bind to receptors on cell membranes to activate the intracellular MAPK signaling pathway, causing an increase in matrix metalloproteinases (MMPs) expression, chondrocyte apoptosis and cartilage destruction. MMPs can degrade almost all the extracellular matrix of chondrocytes and has been shown involved in the pathogenesis of OA (*Murphy et al., 2002*). In addition, MAPK signaling pathway also participates in and regulates chondrocyte enlargement, calcification and proliferation in the process of OA pathology (*Huang et al., 2008*). Therefore, the MAPK signaling pathway can be used as an effective target for the study of new drugs.

Other DEGs PRPRC and PTPN12 are members of the PTP family. On the basis of previous studies, an important part of the pathogenesis of OA is the immune system (*Goldring & Otero, 2011*). The proliferation of T lymphocytes can regulate the immune response, and the biochemical activities of human articular cartilage can directly affect the proliferation of T lymphocytes (*Pereira et al., 2016*). T lymphocytes are lymphoid stem cells derived from bone marrow and distributed to immune systems throughout the body through lymph and blood circulation (*Moro-Garcia et al., 2018*). On the other hand, dendritic cells (DCs) as a crucial component in triggering an immune response, as well as in inflammatory and autoimmune diseases (*Ganguly et al., 2013*). Furthermore, the PTP family is important in the activation of DCs ability to induce T cell activation and function

(*Rhee et al., 2014*). Therefore, the PTP family and its related pathways may provide a new idea to treat OA.

The involvement of B9D1 in OA has not been well studied, but in recent research, mutations of B9D1 result in common skeleton defects (*Yuan & Yang, 2015*). RB9D1 and its related pathways can be used as a supplement to the current traditional treatment of OA.

In brief, four potential key genes in the pathogenesis of OA were identified through bioinformatics screening, including MAPK-14, PTPRC, PTPN12 and B9D1. The imbalance of matrix synthesis and catabolism of OA chondrocytes and the increase of cell membrane destruction may be related to the high expression of MAPK-14 promoting the death of OA chondrocytes. The regulation of the expression of these key genes can provide a potential reference target for the treatment of OA.

## CONCLUSIONS

The hub genes MAPK-14, PTPRC, PTPN12 were upregulated while B9D1 was downregulated after screening from the 265 differentially expressed genes. OA chondrocytes had high expression of COL I protein and low expression of COL II protein. The inflammatory necrosis of OA chondrocytes was increased compared with the normal chondrocytes. The expression levels of MAPK-14, PTPRC and PTPN12 in OA chondrocytes were significantly higher, while the expression levels of B9D1 were significantly lower than that in normal chondrocytes. The MAPK-14 among the four potential key genes has been proved to play an important role in the treatment of OA.

## ACKNOWLEDGEMENTS

We thank Dr. Qian Yi, Dr. Weichao Sun, and Dr. Desheng Sun for their assistance in bioinformatics and cell culture methods.

### Funding

This study was supported by the National Natural Science Foundation of China (81800785, 81972085, 81772394), the Natural Science Foundation of Guangdong Province (2018A0303100027), the Sanming Project of Shenzhen Health and Family Planning Commission (SZSM201612086), Shenzhen Science and Technology Planning (JCYJ20180228163401333), and the Doctor Innovation Project of Shenzhen Health System (SZBC2018015), and Shenzhen Peacock Project (KQTD20170331100838136). The funders had no role in study design, data collection and analysis, decision to publish, or preparation of the manuscript.

### Grant Disclosures

The following grant information was disclosed by the authors:
National Natural Science Foundation of China: 81800785, 81972085, 81772394.

Natural Science Foundation of Guangdong Province: 2018A0303100027.
Sanming Project of Shenzhen Health and Family Planning Commission: SZSM201612086.
Shenzhen Science and Technology Planning: JCYJ20180228163401333.
Shenzhen Peacock Project: KQTD20170331100838136.

## Competing Interests

The authors declare there are no competing interests.

## Author Contributions

- Zhiqin Deng and Xiaotian Hu conceived and designed the experiments, performed the experiments, prepared figures and/or tables, authored or reviewed drafts of the paper, and approved the final draft.
- Murad Alahdal conceived and designed the experiments, performed the experiments, analyzed the data, prepared figures and/or tables, authored or reviewed drafts of the paper, and approved the final draft.
- Jianquan Liu, Zhe Zhao and Xiaoqiang Chen analyzed the data, authored or reviewed drafts of the paper, and approved the final draft.
- Junxiong Xie analyzed the data, prepared figures and/or tables, authored or reviewed drafts of the paper, and approved the final draft.
- Li Duan, Daping Wang and Wencui Li conceived and designed the experiments, authored or reviewed drafts of the paper, and approved the final draft.

## Human Ethics

The following information was supplied relating to ethical approvals (i.e., approving body and any reference numbers):

The study was reviewed and approved by the First Affiliated Hospital of Shenzhen University Health Science Center Research Ethics Committee.

## Data Availability

Raw data are available in the Supplemental Files.

## Supplemental Information

Supplemental information for this article can be found online at http://dx.doi.org/10.7717/peerj.10656#supplemental-information.

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
