# Peer review of "High expression of MAPK-14 promoting the death of chondrocytes is an important signal of osteoarthritis process"

_PeerJ, doi:10.7717/peerj.10656_

## Round 0.1 · original submission · Major Revisions

As you can see, all reviewers raised serious concerns about your study. Therefore, a major revision in needed to address all these critiques and to amend the manuscript accordingly.

Reviewer 1 ·

Basic reporting

The article is professional with structure figures table and raw data shred. It is relevant to the hypothesis.

Experimental design

23 osteoarthritis knee joint samples and 22 samples of normal knee joint were obtained from the microarray omnibus databases. 265 DEGS were identified. MAPK-14, PTPRC, PTPN12 were upregulated while B9D1 was downregulated. Results were confirmed in human OA knee joints for the 4 DEGs. However the conclusion that MAPK-14 is promoting chondrocyte apoptosis was not supported.

Validity of the findings

The findings showed that the microarray data found in the database and the DEGS were verified. However no mechanism or additional new data was presented. It was just a basic confirmation. It is unclear how this will aid int he understanding of OA. Therefore as it stands it is just descriptive suggesting 3 genes are upregulated and 1 is downregulatd during OA.

Reviewer 2 ·

Basic reporting

In the present study, the authors aimed to explore the hub genes in the OA and found that the hub genes MAPK-14, PTPRC, PTPN12 were upregulated while B9D1 was downregulated. Moreover, the inflammatory necrosis of OA chondrocytes was increased compared with the normal chondrocytes.

Experimental design

By bioinformatics, the authors found that the hub genes MAPK-14, PTPRC, PTPN12 were upregulated while B9D1 was downregulated. And qPCR further confirmed the expression of screening differential genes in normal and OA chondrocytes. Compared with normal chondrocytes, OA chondrocytes had high expression of COL I protein and low expression of COL II protein by fluorescence. The inflammatory necrosis of OA chondrocytes was increased compared with the normal chondrocytes by propidiumiodide staining.

Validity of the findings

No comment

Additional comments

The results may provide a novel insight into an signal of osteoarthritis process. However, there are several concerns that should be addressed. It may require major revision before ready for publication.
Major concerns:
(a1)
Overall, there is no evidence provided about the relationship between MAPK14 and the death of chondrocytes. Hence, it would be important to overexpress or interfer MAPK14 in chondrocytes and observe the changes of apoptosis-related markers.
(a2)
It would be more persuasive if the changes of COL I and COL II protein levels in OA and normal chondrocytes are proved by western blot.

Reviewer 3 ·

Basic reporting

A big concern is that there are a lot of grammatical errors and typos in this manuscript. For examples:
- Line 27: ... howerer it is still unclear.
- Line 42: ... the expression levels of B9D1 were significantly lower than that in ...
- Line 56-57: One of the most important risk in the OA development ...
- Line 75: GSE43923 include 3 OA knee joint samples ...
- Line 82: ... users can integrate and analyze multiple datasets for identify DEGs ...
- ...
The authors should re-check and revise carefully. It is better if it is checked by a native speaker or English editing service.

Second concern is the introduction. The authors have not shown clearly the objective/motivation of the work as well as some related literature. Currently, the manuscript includes 2 paragraphs in the introduction without much information to convince that the work is promising and necessary.They should consider rewriting the content of the article in concise language, especially the introduction and method sections, highlighting the key points.

In the abstract, "background" section has not contained much information or key points.

Experimental design

Abbreviation should be defined at the first time use (i.e., OA)

The authors only had few samples in their dataset (23 OA and 22 normal). It is a dataset with very small sample size.

They merged the data from three GEO datasets without considering the inspection and removal of batch effects.

GO database or analysis has been used in previous bioinformatics works such as PMID: 31921391 and PMID: 31277574. Therefore, it is important to refer more works in this description.

Validity of the findings

The concentration of the paper is MAPK-14, but the authors also concluded the findings on COL I, COL II protein, or PTPRC and PTPN12 genes.

The authors should improve the figure's legend to provide more information to readers.

Why did the authors select the confidence score of 0.4 in PPI network?

Additional comments

No comment.

---

## Round 0.2 · accepted · Accept

All critiques were adequately addressed and the manuscript was revised accordingly. Therefore I am happy to accept your manuscript in its present form.